# Parents’ Perceptions of the Value of Children’s Participation in Pediatric Rehabilitation Services: A Phenomenographic Study

**DOI:** 10.3390/ijerph182010948

**Published:** 2021-10-18

**Authors:** Lisa Kronsell, Petra Svedberg, Jens Nygren, Ingrid Larsson

**Affiliations:** Department of Health and Care, School of Health and Welfare, Halmstad University, SE-30118 Halmstad, Sweden; Lisa_kronsell@hotmail.com (L.K.); petra.svedberg@hh.se (P.S.); jens.nygren@hh.se (J.N.)

**Keywords:** child care, disabled children, interviews, parents, participation, phenomenography, value

## Abstract

Ensuring that children have opportunities to be involved in decision-making regarding their own care is associated with quality improvement in pediatric rehabilitation. The aim of the study was to explore parents’ perceptions of the value of children’s participation in pediatric rehabilitation services. Semi-structured interviews were conducted with 17 parents of children with disabilities who visited pediatric rehabilitation services. A phenomenographic analysis method was used. Three categories developed from the analysis describing how participation generated value in terms of empowerment, self-awareness, and independence. The outcome space describes a hierarchical relationship between the categories and their influence on each other. Independence achieved through participation was a core aspect and is at the highest level in the hierarchy since it includes and depends on the outcomes from both empowerment and self-awareness. Parents’ perceptions of the value of children’s participation in pediatric rehabilitation services include the possibility for the child to use their entire capacity through values created in terms of empowerment, self-awareness, and independence, in order to live the best life possible. Children with disabilities are diverse as a group, and further research to identify barriers and facilitators of participation is needed to adjust interventions within pediatric rehabilitation services to ensure that children with disabilities can be increasingly empowered, self-aware, and independent.

## 1. Introduction

The Convention on the Rights of the Child (UNCRC) [1] is of great significance for a child in a vulnerable situation and need of healthcare [2]. Even though the intentions in the UNCRC have largely been incorporated in legislation and regulations and the importance of patient participation has found general acceptance, challenges remain in translating these ambitions into practice [3,4]. Participation within healthcare includes planning for specific activities, training, or treatment [5], as well as the involvement of children in decisions regarding their care [3,4]. Healthcare professionals often fail to provide opportunities for children to understand their health status and conditions, as well as to share their views and to participate in decisions regarding their care [3,4]. For children with disabilities, whose care is characterized by frequent meetings with rehabilitation services, the enabling of their participation by healthcare professionals becomes even more important in terms of safety, care outcomes, and perceived quality. Children who are involved in decision-making feel more prepared and less anxious [6,7], whereas those who are not included in decisions about their care have poorer compliance with treatment recommendations and care plans [4,8]. Ensuring that children have the possibility of being involved in consultations and decision-making regarding their care [4] is therefore highly associated with quality improvement in pediatric rehabilitation.

Children with disabilities usually have an extensive need for support from their parents and healthcare professionals in pediatric rehabilitation services [8]. The parents and the family have important functions in conveying continuity and long-term perspectives and constitute a critical source for support both as part of and beyond the rehabilitation services [9]. The child’s development is dependent on continuous interactions with the family or other healthcare professionals, particularly in early childhood [10]. A family-centered care approach could thus be of great importance for children with disabilities [9]. It includes the perspectives of both children and parents [4,11] and ensures that the child receives care in the context of their family and community. This entails that the care and decision-making reflect the child within their family, home, school, daily activities, and quality of life [12]. This approach also recognizes the need for the child to develop capabilities for a future life without the presence of their parents. This is especially important in a context where children with disabilities have special needs in order to have the capacity to be fully participating.

Parents often have insights regarding their child’s strengths and disadvantages, and they can emphasize more aspects that are important for helping the child become more involved in their rehabilitation with the purpose of optimizing the value for the child [2,13]. The parents’ perspectives of which outcomes there are from their child’s participation can emphasize the value for the child in a long-term perspective and lead to better ways of developing healthcare for their child. The definition of value in healthcare has not gained universal agreement, but patients have identified communication, healthcare access, and shared decision-making as key elements in a value-generating healthcare environment [14]. The understanding of value can be highly personal and dependent on the context, resulting in what is valuable for one person not necessarily being valuable for the other [15]. Healthcare can be viewed as a service, with consumers and providers of services working together to coproduce valuable outcomes [16]. Gaining knowledge of parents’ perceptions of the value of children’s participation in pediatric rehabilitation services is needed to help children and health professionals develop the necessary competencies for effective partnership as not all patients have the capacity to be active participants in co-producing their healthcare service [16]. The coproduction of value in a family-centered care approach might be even more important in healthcare involving children with disabilities due to its continuous and life-long nature. This highlights the need to further explore parents’ perceptions of the value of children’s participation in pediatric rehabilitation services to create the necessary conditions for the children. A previous study, focusing on developing and implementing a digital decision support tool to increase the participation of children with disabilities in pediatric rehabilitation, emphasized the importance of inviting children to share their needs [8]. The study brought new knowledge as to how this specific patient group viewed participation. However, research has primarily focused on how participation is carried out, not the value being generated through participation. Isolating children’s perceptions from parents’ perceptions might thus be inconclusive when researching children’s participation within pediatric habilitation services. The parents’ perceptions are therefore important in order to gain greater knowledge of the value generated in children’s participation in pediatric rehabilitation services, and the aim of this study was to explore variations of parents’ perceptions of the value generated in children’s participation in pediatric rehabilitation services.

## 2. Materials and Methods

### 2.1. Design

A phenomenographic approach was chosen to describe and understand the different ways a group of people understands a phenomenon [17] and is frequently used in healthcare services research [18]. Since phenomenography involves different ways of understanding, both the “what” aspect, which tells us what the focus for the subject is, and the “how” aspect, which describes how meaning is created for the person, is included in this study approach [19]. A phenomenographic approach was thus chosen as it aims to describe the variations in how parents perceived the value of their children’s participation in pediatric rehabilitation services. The COREQ checklist for qualitative studies was used to assure quality standards [20].

### 2.2. Participants

The target group was parents of children with physical disabilities, intellectual disabilities, or autism spectrum disorders that visited pediatric rehabilitation services. A purposeful sample of interviews was conducted with 17 parents, of whom 13 were mothers and 4 fathers (Table 1). Parents of children with various disabilities were included in the study in order to gain a variety of perceptions of the phenomenon being studied.

### 2.3. Data Collection

Data were collected through individual interviews. The interviews took place between October 2017 and April 2018 and were held by the researcher (I.L.). The interview started with the researcher clarifying the aim of the study. A semi-structured interview guide with open questions focusing on children’s participation in pediatric rehabilitation was used. Examples of questions were “Can you describe your child’s participation in pediatric rehabilitation?”, “What does it mean for your child to be involved in pediatric rehabilitation?”, “What can influence your child’s ability to participate in pediatric rehabilitation?”, “What do you think about children’s participation in pediatric rehabilitation?” The participants were encouraged to provide more in-depth information by asking them to “explain more” or with questions such as “how do you mean?” or “what do you think of when you say…?” Each interview was performed by telephone and was audio-recorded. The interviews lasted between 32 and 103 min, with a median of 59 min. The total interview time was 17 h and 7 min, and the interviews were transcribed verbatim.

### 2.4. Data Analysis

The analysis was carried out according to the procedure of Larsson and Holmström [19], aiming to identify various ways of parents’ understanding the value of their children’s participation in pediatric rehabilitation services. The analysis started with the reading of all the interviews followed by a second reading while extracting answers connected to the aim of the study. All extracted meanings were marked with the given number of the interview to obtain structure and transparency. The third step was to look for the focus of each parents’ attention and how they described the value of their child’s participation in rehabilitation. A preliminary description of each parent’s predominant perception was made, and the descriptions thereafter merged into categories based on similarities and differences. These mergences composed a total of three descriptive categories, which were formulated and reported in the form of text and quotes (Table 2). The identified categories were as follows: participation generates value in terms of empowerment, participation generates value in terms of self-awareness, and participation generates value in terms of independence. Identifying non-dominant perceptions, which constituted the fifth step, was performed to ensure that no aspect was overlooked (Table 3). The final step consisted of structuring the outcome space. The analysis was carried out by the first author while having regular discussions with the other co-authors who have extensive experience in healthcare research and qualitative methodology.

### 2.5. Ethical Considerations

The study was approved by the Regional Ethical Review Board at Lund’s University, Sweden (no. 2017/707). The study conforms to the ethical principles as set out by both the National Guidelines [21], as well as the ethical principles for research on human beings by the World Medical Association in the Declaration of Helsinki [22]. The participants received written and verbal information about the study and provided written consent to participate. They chose the time for the interviews and were all informed about the voluntary of the study, confidentiality, and the possibility to withdraw at any time. All participants were offered the possibility to discuss any emotions or thoughts that may have occurred during the interview.

## 3. Results

The parents spoke of several ways of understanding the value of children’s participation in pediatric rehabilitation services. The analysis resulted in three categories, in which emphasis was set on values concerning the children’s present and future. The categories were as follows: participation generates value in terms of empowerment, participation generates value in terms of self-awareness, and participation generates value in terms of independence (Table 4). Quotations from the participants’ descriptions were used to provide illustrative examples.

### 3.1. Participation Generates Value in Terms of Empowerment

The category participation generates value in terms of empowerment consists of the sub-categories: to be involved, to be respected, and to have self-esteem. Participation in pediatric rehabilitation services in terms of being involved in decisions and activities ensures that children know they are respected and have the right to control their life, which in turn creates opportunities for these children to experience empowerment. Participation generates value in terms of empowerment directly but also for the future. The children can feel in control over their lives and more positive to overcoming difficulties by developing greater self-esteem and knowledge about their rights.

#### 3.1.1. To Be Involved

Being involved generated value in terms of empowerment by feeling in control of life choices and everyday life. This included decisions within the pediatric rehabilitation services and was also described as increasing the power the children would have to impact their future in demanding participation. To be involved thus ensured that the children were granted access to their care, that they felt content with their prospects and what they wanted to accomplish in life.


*“That I am an individual, I can go my own way in life. No one else decides how my life should be, it is me who decides how my life should be, for better or for worse. If I can take part and decide, it could be easier to be what I want in my life.”*
(Parent no. 5)

The extent of their involvement would vary depending on the child’s disability. Having a structure in the form of preparations was essential for contributing to the children’s involvement in their rehabilitation, and following routines established a safe environment and was described as a useful way to enable involvement for the children. Creating opportunities for the children to experience empowerment thus consisted of involvement in both preparing and performing activities.


*“He likes to know a long time ahead what’s going to happen… It is very important for him that you really do what has been planned and that he’s included in the planning, how he wants things.”*
(Parent no. 6)

#### 3.1.2. To Be Respected

Value in terms of empowerment was generated by ensuring the children were treated with respect, with their disability or difficulties being considered. One way to achieve this was by providing equal opportunities for participation in activities, by using assistance. Other ways to ensure a respectful encounter were for professionals to address the children directly, not neglecting them in conversations or decision making. However, parents would sometimes be an important part of a respectful conversation. True respect for the children entailed ensuring they had suitable help and assistance to enable their participation, which for example could be established with the support of the parents as a translator.


*“… it was terrible in that place where we were. They should know that these children have different ways and techniques of communicating. The person who’s going to perform that type of examination should be required to be able to draw. Not just have an education in psychology but also be able to meet these children where they are. Maybe adjust the material they use for the examination in the way that is needed for evaluating the child in a fair way.”*
(Parent no. 10)

Tailor-made inventions, adjusted treatments, and support allow respectful and equal participation. Ensuring professionals understand the children’s disabilities, which include both difficulties as well as strengths, was perceived as essential for the children to be respected. Working from these criteria ensures the children feel respected and have the opportunity to experience empowerment by knowing their rights, not only within the pediatric rehabilitation services but in other areas of life as well.


*“Then you also notice how fantastic it is when a child ends up in the right environment. With the right tools and how much it benefits a child’s well-being.”*
(Parent no. 13)

#### 3.1.3. To Have Self-Esteem

Having a high level of self-esteem was described as benefitting the children as they felt themselves valued in knowing that their opinions and emotions were important. Building self-esteem was described as a process. Participating by expressing opinions within the pediatric rehabilitation services was perceived as a practice for future situations if the children needed to stand up for themselves in terms of their disability. This contributed to a higher level of self-esteem and created opportunities for the children to experience empowerment as it emphasized the children’s intrinsic worth.


*“If he has a higher level of self-esteem it becomes easier for him to step into adulthood when I leave him. When it comes to work, friends and everything around him. He’s going to stand up without me and build his own life, without me.”*
(Parent no. 16)

Experiencing success boosts the children in believing in themselves, which positively affects the results of the activity or training. Accomplishing tasks was important for the children’s self-esteem and self-confidence. Parents wanted their children to be strengthened in feeling they could try again if they failed instead of giving up. A high level of self-esteem could thus contribute to better outcomes and was perceived as an advantage since it made the children less afraid of experiencing failure. As participation contributed to a higher level of self-esteem, it generated value in terms of empowerment as the children had faith in their own abilities and were able to focus on their strengths.


*“Earlier she said I can’t do anything; I’m just stupid. She felt that she was never allowed to dazzle or be good at something or show something. Then they’d only take her to another room at the other school. There they didn’t think she understood or could participate. She felt very excluded.”*
(Parent no. 13)

### 3.2. Participation Generates Value in Terms of Self-Awareness

The category, participation generates value in terms of self-awareness consists of two sub-categories, to be aware and to be motivated. The participation brought value to the children by allowing them to become more aware of their disability and how it affected them. Using the children’s interests as motivation to participate within the rehabilitation services increased the children’s willingness to participate. Developing self-awareness was valuable both for the children’s present as well as for their future and supported them in continuing with activities they found motivating. Self-awareness was perceived to benefit the children’s rehabilitation outcomes by acknowledging the children’s present and future concerns and also using the children’s preferences in the rehabilitation.

#### 3.2.1. To Be Aware

Parents perceived participation in terms of the children learning about their disability that leads to self-awareness. Awareness of how their disability affected them, physically and mentally, and understanding their need to visit the pediatric rehabilitation services was perceived as a requirement for the children to participate. By expressing feelings and worries about the future the children could begin the process of accepting their disability to strengthen their mental health and being able to live the best life possible. Self-awareness was stimulated when the children’s concerns of future abilities or feelings of distress were acknowledged in the pediatric rehabilitation services.


*“There’s some sort of grief for him. It’s because one has the difficulties one has. He hasn’t put so much energy into it. Because it’s something he’s always lived with. It’s nothing that has appeared overnight. At the same time, one needs some help to tackle those feelings that arise about it.”*
(Parent no. 6)

The children’s participation was valuable as it contributed to them being aware of their disability and made it possible for them to identify the assistance they needed. Being aware also opened up the possibility to learn how different people behave in social situations, which was important for the development of the children’s social relationships. Understanding emotions and feelings could stimulate self-awareness and an understanding of oneself as well as of other people. This facilitated the children’s social interactions as it made them aware of their disability, enabling them to explain their needs to other people.


*“She can’t always use the strategies she wants to use when things go a little wrong. She walks away and is allowed to be alone for a while. If it’s something that’s going against her and doesn’t go according to her will. She has learned to cope with that.”*
(Parent no. 15)

#### 3.2.2. To Be Motivated

Explaining and ensuring the children understood the advantages by participating was perceived as a key aspect of motivation. The children persisted with the training when it resulted in outcomes that were important to them. Motivation generated a determination for the children, as they recognized the change participation could make in their life. This generated value in terms of self-awareness since the children became aware of the importance of continuing with their rehabilitation.


*“When she’s into it and thinks it’s fun, she then lights up like a ray of sunshine and becomes social. Then the exercises go often much better.”*
(Parent no. 12)

Using activities that the children did not experience as training was perceived as motivating, such as when exercise was not the main focus of the training but still performed. The professionals’ ability to motivate the children, knowing their interests and desires to do so, were thus important. The children were motivated to accomplish their assignments when their preferences were included in the rehabilitation program, which also created a sense of meaningfulness and self-awareness. Parents perceived that the will to participate was strengthened and the outcomes from the rehabilitation improved by letting the children express their wishes and interests. Self-awareness thus developed as the children were given the chance to ask questions about their disability and were given explanations as to why they had to perform certain activities. Being aware of the expected outcomes of their participation were motivating.


*“Yes, it might benefit the child more. I think it might be a little more exciting to go to these things. When one more looks forward to something instead of just feeling ‘oh, do I have to go there again’.”*
(Parent no. 9)

### 3.3. Participation Generates Value in Terms of Independence

This category, participation generates value in terms of independence, consists of the sub-categories to be understood when communicating, to have self-determination, and to practice abilities. The value of participation was to ensure the children would be able to live an independent life in the future since the parents would not be able to continue to take care of their adult child the same way. Value was found in the possibility for answering questions and being understood when communicating, and thus choosing which information to share with others, ensuring the child’s integrity through self-determination. Parents also emphasized that the level of independence was dependent on the children’s capabilities, within the rehabilitation as well as all aspects of the child’s everyday life. Practicing their abilities was an important way of increasing independence. Generating value in terms of independence was perceived as a process where the children would learn how to be independent in their current life situations, which would also influence their future opportunities for an independent life.

#### 3.3.1. To Be Understood When Communicating

Children with disabilities have different needs in their communication that can affect how they are understood. Some children communicate via speech, via sign language, or with the assistance of visual language. Parents emphasized how the possibility of communicating, in ways that other people would understand, was an essential part of creating independence. To be able to communicate with the assistance of communication aids without being dependent on a family member created opportunities to increase the children’s independence in several ways. Having a communication tool, e.g., a mobile phone, could make it easier for the children to raise sensitive topics that they otherwise would have been uncomfortable to talk about. Using a mobile phone as a communication tool also made the children feel more liable to use it in their everyday life as it was a very common device among other children as well. This accessibility could therefore increase the participation and independence of the children.


*“ … and that they understand him. Because if someone doesn’t understand him, it is difficult … one might feel depressed.“*
(Parent no. 8)

Encouraging the children in their communication form was described as a learning process. The children would practice expressing needs and emotions, sometimes with the support of a parent to eventually manage the communication on their own in the encounters with healthcare professionals in their pediatric rehabilitation. This was perceived as being a great help for the child at his or her level of independence concerning possibilities of formulating how they feel and asking for the help they need. One parent expressed it with the following words: “A way to make it easier to be involved in decision making, even if you don’t have a spoken language”. (Parent no. 5)


*“You can immediately see that she becomes sad and doesn’t like this standing aid. Because she’s in such pain. But it’s perhaps not always so easy for the habilitation professionals to understand her then, what she means. Then I’ve gone in and explained that she gets cramps, in her leg, in her calves. When it’s too tight.”*
(3)

Being able to communicate helped the children to express themselves to people in their surroundings without the risk of being misunderstood. It was perceived as a security for the children to rely on their ability to make themselves understood. Communication was valuable in other aspects, such as in school with other children in order to be independent. The parents described how communication played a part in their children’s social life, and that not being able to communicate with other people contributed to alienation and sadness, which could result in the child not wanting to try to communicate or speak at all.


*“It’s one of those examples where she only wanted to say that she was in pain, but she couldn’t say it. Because then the teacher said, why didn’t you say so (states name), and then she just stood there being very sad. Then they had started to scream and shout at her. She immediately felt that it wasn’t worthwhile, she wouldn’t be understood.”*
(Parent no. 13)

#### 3.3.2. To Have Self-Determination

Being self-determined was perceived as important for the children’s independence and it included being responsible for choices that affected their life and well-being. Allowing the children to be responsible for choices affecting their life within the rehabilitation service prepared them for an independent future. The children’s self-determination increased as a result of knowing they had the possibility and right to participate without the presence of a parent.


*“They’ll do a survey and then formulate goal. She’s done this herself in the last years. Then it’s really her wishes that are included. Which goals she wants to attain, which we might have discussed a little at home earlier. She has told us, and we have been able to say what we think. But it is she who decides when she’s in there.”*
(Parent no. 11)

The children were responsible for their encounters with the healthcare professionals in their rehabilitation, which also prepared them for an independent life in the future. The parents described how self-determination ensured the children’s integrity as they could participate on their own terms. This also included the right for the children to say no.


*“For example, we have therapy sessions now, which we think is important. So he’ll be able to talk to someone in a place where he knows he can say whatever he wants. Without his mom or dad being present and having opinions or can hear it.”*
(Parent no. 2)

#### 3.3.3. To Practice Abilities

Practicing abilities was perceived as important for the children’s possibilities of experiencing independence. Practicing includes exercise in terms of being physically active, for maintaining or developing the child’s body functions. The process of practicing abilities was therefore influencing the children’s life as it sustained the possibility of being independent and also contributed to facilitating situations in the future in which the children were currently experiencing difficulties. The pediatric rehabilitation services constituted a safe place for the children to practice their abilities in proper circumstances. Practicing abilities was perceived as creating possibilities for the children to develop problem-solving and thus experience independence.


*“He has a great deal of trouble with his hands, but it works, and he feels he can function in his everyday life. But you try to adjust as much as you can at home to make it easy for him. From pouring things into smaller containers, to see that the cereals are always in a jar where the lid is easy to take off.”*
(Parent no. 7)

Learning to be responsible was also part of the parents’ description of the practice of being valuable for the children. Being allowed to take responsibility for following necessary routines created possibilities for the children to develop independence. Practicing managing hands-on activities, such as taking medication, was affecting the children’s possibilities for independence. The future value of independence was connected to the child’s progress in learning how to take responsibility without the assistance of a parent or a caregiver. The parents perceived the present and future aspects of independence as important for the well-being of both children and parents.


*“You could see there was a big difference when she suddenly got to be responsible for those kinds of things herself. We had spoiled her a bit too much there, and that’s something we as parents have also done. But we’ve started to let go, within the rehabilitation. Then she has grown.”*
(Parent no. 11)

Practicing abilities also included the children’s possibility of developing their mental and social abilities in order to improve their interactions with people outside the family. The aspect of practicing social abilities was thus perceived as specifically important for the children’s future independence.


*“That’s where we want her to get on, to practice the social aspects. They do that a lot there and we look forward to it. Because I feel deeply worried about the future. That NN will isolate herself completely. That she’s stuck in an apartment.”*
(Parent no. 17)

### 3.4. Outcome Space

The three categories of understanding and the internal relationships constitute the outcome space (Figure 1) and have been interpreted to represent the variation of the parents’ collective understanding of the value of participation and not the individual variation between parents. The categories are hierarchically related. The categories participation generates value in terms of empowerment and participation generates value in terms of self-awareness are situated on the same level in the hierarchy since they reflect two essential ways of understanding the value being generated for the children by participation. The category participation generates value in terms of independence is on a higher level of the hierarchy and includes the advantages of the other two categories. In analyzing the outcome space, the value in terms of empowerment included control over one’s life, respectful treatment, and developed self-esteem, which relates to value in terms of independence since empowerment can be viewed as a means of creating possibilities for independence. Independence also includes decision making where the value in terms of self-awareness shows that a greater ability to make independent decisions desirably influences one’s life. Participation generates value in terms of independence reflected the most comprehensive way in which the parents understood the value of participation and thus constitutes a core value. This shows that the parents’ perception of independence for their children relates to both empowerment and self-awareness as a process towards independence. However, the categories are still connected as they influence different aspects of each other within the hierarchical relation. As empowerment can be seen to benefit independence, independence can also influence the opportunities for experiencing empowerment when communication, self-determination, and practice are encouraged. The categories are therefore linked to one another in different ways depending on the context of the children’s participation.

## 4. Discussion

This study aimed to explore the value of children’s participation in pediatric rehabilitation from the parents’ perspectives. The study showed that parents understood the value created through participation for children in pediatric rehabilitation services in three different ways: value generated in terms of empowerment, self-awareness, and independence. Giving the children possibilities to participate in every aspect of their rehabilitation was perceived as important in generating these values. Being involved, respected, and developing self-esteem generates opportunities for experiencing empowerment. Experiencing empowerment creates possibilities for the child to be in control of both present and future aspects of life. Being aware and motivated leads to the children’s self-awareness now and in the future. Participation within the pediatric rehabilitation service thus influences the child’s well-being in both immediate and long-term perspectives. Learning how to communicate, practice abilities, and be self-determined within a safe environment creates opportunities for the children to become independent. The children thus become more prepared for a life less dependent on pediatric rehabilitation services. Increased participation for disabled children may affect the child’s health in several positive ways and contribute to increased empowerment and independence [23]. The results show that the value generated through participation could make it possible for children to use their full potential and help them to become all that they are capable of being. The parents’ perceptions contributed to revealing that children with disabilities might need extra support in understanding prerequisites and develop abilities. An aspect of particular interest, which would not have been considered without the input from the parents, is the view of participation as a process in understanding the generation of value.

The result reveals that parents perceived the value generated through participation in terms of empowerment, in which the children are given the power to make decisions and to be in control of life to a greater extent. This relates to research defining empowerment as having control over the determinants of one’s quality of life, where health is a determinant of quality of life and is seen as a resource in everyday life [24]. Viewed as a process through which the children gain greater control over decisions and actions influencing their health and lifestyle [25], empowerment strengthens the children’s possibilities to be involved, respected, and self-determined the more they participate. The parents described that this was important for the children’s future possibilities and capabilities, which relates to empowerment having both a terminal and an instrumental value. It shows the importance of professionals’ capabilities of letting the children have or acquire as much control over the change process as possible [26]. Previous research has concluded that healthcare professionals often fail to provide opportunities for children to understand their conditions, share their views, and participate in decisions regarding their care [3,4]. A newly published study [27] reveals that the significant obstacles for child participation are that the rehabilitation services are adult-centered and inflexible with norms of non-participation. A conclusion from this study, based on the perspectives of children and young people, is that there is a risk that norms are being reinforced unless children are allowed and educated to participate from an early age [27]. Young people have described services to be helpful when they are individualized and flexible in the amount and type of support according to current needs [28]. Children with disabilities, who are in a vulnerable situation, need extra support within pediatric rehabilitation [8]. The result shows that this need can be avoided by increasing their participation to reach a level of involvement that is built on respect towards the child and confidence in their own capabilities. This also strengthens the need to incorporate the family-centered care approach as this allows the parents to be present and observe situations where the prerequisites for the child’s participation are not living up to required needs. As the family-centered care approach is consistent with the ideals of empowerment, respect for personal autonomy, and human rights [29], it emphasizes important aspects of care that are associated with promoting the child’s participation in the care process and thereby the generation of value for the child through the care.

Participation generated value for the children in terms of self-awareness as they became aware of their disability and became motivated when their preferences were attended to. This supports the idea that children who are not included in decisions about their care may be less willing to follow treatment recommendations and care plans [4,8]. Thus, from a child perspective, meetings with healthcare professionals need to be adapted to them and include alternative and augmentative communication tools in order to be engaging and motivating [27]. When a person’s motivation is autonomous, they experience better behavioral outcomes as well as improved wellbeing and mental health [30]. This shows the importance of an individual care plan where the child’s interest is included for the child to attain desirable outcomes [24]. The results show that the value in terms of self-awareness includes having proper knowledge for making decisions that benefit one’s health. The children’s possibilities of making decisions that will benefit their future wellbeing are improved by being aware of and understanding their disability, as well as having the motivation to develop in areas they consider important. This relates to shared decision making (SDM) and the importance of understanding one’s care in order to be able to be involved in decision making. Findings from a scoping review [31] showed that facilitators for SDM in pediatric care are high-quality information that is tailored to the child’s developmental and literacy needs. From a child’s perspective, if they feel that they are listen to and if the healthcare professionals ask for their opinions, feelings of influence and power emerge [32]. Children thus need accessible knowledge about themselves and why they are attending the rehabilitation service, something previous studies have revealed as deficient within pediatric care [3,4].

Independence for the children constituted a core value of participation in this study as it reflected the most comprehensive way in which the parents understood the phenomenon. It consisted of communication, self-determination, and getting practice, which was perceived as essential for the children’s future when their parents would not be there for them in similar ways as today. Independence can be viewed as taking responsibility for one’s health [24]. While self-awareness can make it possible for the children to gain knowledge of which decisions benefit their health, independence allows the children to take responsibility for their health and lifestyle. As stated earlier, independence was also viewed as a process within participation, where the possibility to practice abilities was a part of the learning process of being independent. Although parents can provide support in detecting barriers for the children’s participation by being present at rehabilitation meetings, some parents expressed the importance of not stepping in and allowing the children to practice how to be more independent and develop their capacity to take more responsibility for their rehabilitation. This relates to other research describing how parents both serve as facilitators and barriers for their child’s possibility to participate [8,27,33]. It also relates to the importance of the settings and the healthcare professionals’ approaches towards the child’s participation and independence from their parent, for the children’s participation to take place [8,33]. Gaining independence for the children can involve important aspects for the parents as well [2], especially in situations where the children have difficulties expressing themselves through speech [34].

The ability to communicate and be understood can increase a sense of security for both children and parents. Recognizing the importance of communication leads to the risk that children are afraid of trying to express themselves if they fear being misunderstood. Allowing for and expecting the child to be more involved in communication could therefore entail a risk for increasing their feelings of being excluded [33], which could be a barrier to their own independence. Supporting children’s communication is, therefore, crucial for them to become active agents within their rehabilitation. Appropriate communication tools that support the awareness of the child’s perspective can address the challenges of enabling communication within pediatric rehabilitation and provide new ways of communication for children with disabilities. Parents perceived a digital communication tool in this study, for example, a mobile phone, as contributing to accessible communication in all areas of children’s life. Previous findings have shown that using a digital communication tool provided children with the possibility of being heard while being silent [7,35]. Using existing knowledge to develop digital communication tools to modify participation for children who desire less interaction through talking can thus create further possibilities to invite children to participate. Digital tools within pediatric rehabilitation might also open up for developing possibilities for remote participation as well. The findings in this study support previous research demonstrating that parents felt a sense of relief and were able to take a step back as their child gained greater independence [2,6]. The results further support the need to explore parents’ perceptions of the value of participation in relation to children’s experiences, since the understanding of the phenomenon is complex, and a variety of perceptions contribute to a more nuanced understanding of how to view the value of children’s participation overall. Children have spoken of independence where physical and cognitive access are concerned in previous studies [8], but a future perspective of independence has been missing, which the parents in this study have emphasized.

### 4.1. Methodological Considerations

Trustworthiness in qualitative inquiry is determined by the four criteria: credibility, dependability, confirmability, and transferability [36]. Credibility was strengthened through rigorous discussions between the first author and the other co-authors during analyses while bearing in mind preunderstanding, as well as a thorough description of the findings. A phenomenographic approach aims to identify variations, and credibility was strengthened in this study through the diversity of participants [19] ensuring variation in ways the parents were perceiving and conceptualizing the same phenomena. Dependability refers to the stability of the data [36]. Dependability was strengthened by using the same open-ended questions to all the participants to assist them to reflect and explain their understanding of the phenomenon of the value of participation. The strength was that the interviews were all performed in the same way. The weakness was that the interviewer gained new insight into the phenomenon throughout the interviews, which might have influenced the follow-up questions. Looking for both the dominant and non-dominant perceptions ensured that a multitude of possible ways of describing value was revealed. Confirmability was demonstrated by the systematic and conscientious data analysis, where all steps have been reported. Quotations of the parents’ ways of understanding were presented, which enhanced and illustrated the results. This also allows the reader to judge the interpretation and enabled transparency of the analysis. A qualitative approach does not attempt to generalize the result to a whole population [19], but as the target group was diverse and included parents of children with various disabilities, the perceptions of the value of participation could be transferable to other children with disabilities.

### 4.2. Implications

Children with disabilities are a diverse group and further research to identify specific barriers and facilitators of participation is needed. Such research could contribute to adjusted interventions within pediatric rehabilitation services as well as other environments where the children reside, in order to ensure that children with disabilities can be empowered, self-aware, and independent in relation to their prerequisites. The results support the need for new methods to verify that participation is adjusted to each child’s ability and will. Digital tools for participation are one example of future possibilities for supporting participation according to the child’s preferences and creating possibilities for remote participation. The study implies that this could lead to better living conditions for the children, which influences their future life and health. Developing and implementing these methods may generate difficulties, but the present study shows that parents can have a facilitating role within the children’s rehabilitation through their knowledge about how to attract and enable participation for their child. Exploring children’s perceptions of the value of their participation in pediatric rehabilitation might contribute to increased satisfaction and quality of care and relate to the purpose of a family-centered care approach to give the best possible care to the child within his or her situation.

## 5. Conclusions

Parents’ perceptions of the value of participation for children in pediatric rehabilitation services include the possibility for the child to use their entire capacity through empowerment, self-awareness, and independence in order to live the best life possible. Empowerment by being involved in decisions strengthens the children’s knowledge of their right to participate and have access to their care while being met with respect. It also affected the children’s self-esteem when their worth and their capabilities were emphasized. Ensuring the children have the possibility of experiencing awareness and motivation can contribute to their self-awareness and to further possibilities of making decisions that can positively affect their health. The advantages of the value of empowerment and self-awareness constitute the foundation of the value of independence. Independence was an important aspect for children to be able to live a meaningful life and was a core value in the parents’ perceptions of the value of children’s participation in pediatric rehabilitation.

## Figures and Tables

**Figure 1 ijerph-18-10948-f001:**
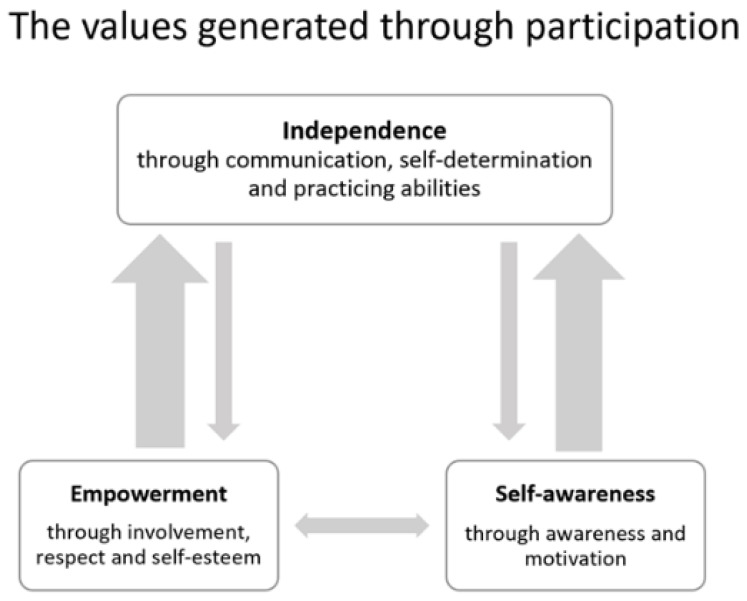
The outcome space—the parents’ collective understanding of the value of children’s participation in pediatric rehabilitation services and the relationship between the categories.

**Table 1 ijerph-18-10948-t001:** Sociodemographic data of the participating parents to children attending pediatric rehabilitation services (*n* = 17).

Variable	Parents (*n* = 17)
**Sex** (*n*)	
Female	13
Male	4
**Age years** median (range)	43 (31–62)
**Civil Status** (*n*)	
Co-habiting	12
Living alone	5
**Education Level** (*n*)	
Primary school	0
Secondary	8
University	9
**Employment** (*n*)	
Employed full time	7
Employed part-time	9
Parental leave	1
**Native-born**	15
**Foreign-born**	2
**Experiences of pediatric rehabilitation (years)** median (range)	5.5 (0.1–16)
**Age of the child in pediatric rehabilitation (years)** median (range)	13 (6–16)
**Gender of the child**	
Female	8
Male	9
**Children’s main disabilities**	
Physical disability	6
Intellectual disability	7
Autism spectrum disorder	4

**Table 2 ijerph-18-10948-t002:** Example of how the analysis was conducted.

Extracted Statements	Code	Sub-Category	Category
That I’m an individual, I can go my own way in life. No one else decides how my life should be, it is me who decides how my life should be, for better or for worse. If I can take part and decide, it would be easier to be what I want in my life.	Being able to affect your life	To be involved	Participation generates value in terms of empowerment
Since she can’t give all the answers she’s expected to, they ask me as if she didn’t exist… If she notices that kind of attitude, she doesn’t answer at all.	Treatment based on the child’s conditions	To be respected	
That you believe in yourself and that you can succeed with several things. I want to do this, but I am in a wheelchair and will manage it anyway. Quite simply, a belief in oneself.	Believing in oneself	To have self-esteem	

**Table 3 ijerph-18-10948-t003:** The parents’ ways of understanding the value of participation for children within pediatric rehabilitation services. Predominant (++) and non-predominant (+) ways.

Participant Number	Sex	Participation Generates Value in Terms of Empowerment	Participation Generates Value in Terms of Self-Awareness	Participation Generates Value in Terms of Independence
1	Female	+	+	++
2	Male	++	+	+
3	Female	+	+	++
4	Male	+		++
5	Female	++	+	+
6	Female	+	++	+
7	Female	+	+	++
8	Female	+	+	++
9	Male	+	++	+
10	Female	++	+	+
11	Female	+	+	++
12	Male	+	++	+
13	Female	++	+	+
14	Female	++	+	+
15	Female	+	++	+
16	Female	+	+	++
17	Female	+	+	++

**Table 4 ijerph-18-10948-t004:** Overview of categories and sub-categories describing parents’ perceptions of the value of children’s participation in pediatric rehabilitation services.

Categories	Participation Generates Value in Terms of Empowerment	Participation Generates Value in Terms of Self-Awareness	Participation Generates Value in Terms of Independence
Sub-categories	To be involved	To be aware	To be understood when communicating
	To be respected	To be motivated	To have self-determination
	To have self-esteem		To practice abilities

## Data Availability

Not applicable. The data will not be shared as ethics approval for the study requires that the transcribed interviews are kept in locked files, accessible only to the researchers.

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
