# Peer review of "Parents’ Perceptions of the Value of Children’s Participation in Pediatric Rehabilitation Services: A Phenomenographic Study"

_ijerph, 2021, doi:10.3390/ijerph182010948_

Round 1
Reviewer 1 Report
This study is judged to be a very good study of parents' perception of children's participation in the decision-making process of pediatric rehabilitation. Until now, children in pediatric rehabilitation have been considered only object, not subjects who actively participate in treatment.
From this point of view, I think the result "Parents' perceptions of the value of children's participation in pediatric rehabilitation services include the possibility for the child to use their entire capacity through values created in terms of empowerment, self-awareness, and independence, to live the best life possible" presented in this study is a very meaningful result.
Thank you very much for the opportunity to review your very interesting work.
Author Response
Comments to the Author
This study is judged to be a very good study of parents' perception of children's participation in the decision-making process of pediatric rehabilitation. Until now, children in pediatric rehabilitation have been considered only object, not subjects who actively participate in treatment.
From this point of view, I think the result "Parents' perceptions of the value of children's participation in pediatric rehabilitation services include the possibility for the child to use their entire capacity through values created in terms of empowerment, self-awareness, and independence, to live the best life possible" presented in this study is a very meaningful result.
Thank you very much for the opportunity to review your very interesting work.
Answer: Thank you
Reviewer 2 Report
The result are very general. At least in the discussion, the authors could cite more specific studies that point to improvements in the motor area of these children and their involvement in practical life. There are certainly studies that quantify the relationship between rehabilitation and the involvement of disabled children in life.
Author Response
The result are very general. At least in the discussion, the authors could cite more specific studies that point to improvements in the motor area of these children and their involvement in practical life. There are certainly studies that quantify the relationship between rehabilitation and the involvement of disabled children in life.
Answer: Thank you for drawing our attention to this. We have now added 5 new references linked to the target group
Extensive editing of English language and style required
Answer: A native Englishman, working as a professor in nursing and thereby knowing the context, has edited the English language
Reviewer 3 Report
This paper is aiming to explore the significance (value) of children’s participation in pediatric rehabilitation from their parents’ aspect. Concretely, using a phenomographic analysis for the interviews with parents, Empowerment, Self-awareness and Independence are extracted as three categorical values of children’s participation in pediatric rehabilitation, and their hierarchical relationships are clarified. Among those, Independence is on a higher level of the hierarchy, and is recognized to be very important for children with disabilities to live a meaningful life in the future.
This paper is sufficiently original and practically useful.
Author Response
This paper is aiming to explore the significance (value) of children’s participation in pediatric rehabilitation from their parents’ aspect. Concretely, using a phenomographic analysis for the interviews with parents, Empowerment, Self-awareness and Independence are extracted as three categorical values of children’s participation in pediatric rehabilitation, and their hierarchical relationships are clarified. Among those, Independence is on a higher level of the hierarchy, and is recognized to be very important for children with disabilities to live a meaningful life in the future.
This paper is sufficiently original and practically useful.
Answer: Thank you
Moderate English changes required
Answer: A native Englishman, working as a professor in nursing and thereby knowing the context, has edited the English language